# Evaporation-induced sintering of liquid metal droplets with biological nanofibrils for flexible conductivity and responsive actuation

Xiankai Li[1,2], Mingjie Li [ID][1], Jie Xu[1,2], Jun You[1], Zhiqin Yang[3] & Chaoxu Li[1,2]

Liquid metal (LM) droplets show the superiority in coalescing into integral liquid conductors applicable in flexible and deformable electronics. However, the large surface tension, oxide shells and poor compatibility with most other materials may prevent spontaneous coalescence of LM droplets and/or hybridisation into composites, unless external interventions (e.g., shear and laser) are applied. Here, we show that biological nanofibrils (NFs; including cellulose, silk fibroin and amyloid) enable evaporation-induced sintering of LM droplets under ambient conditions into conductive coating on diverse substrates and free-standing films. The resultants possess an insulating NFs-rich layer and a conductive LM-rich layer, offering flexibility, high reflectivity, stretchable conductivity, electromagnetic shielding, degradability and rapid actuating behaviours. Thus this sintering approach not only extends fundamental knowledge about sintering LM droplets, but also starts a new scenario of producing flexible coating and free-standing composites with flexibility, conductivity, sustainability and degradability, and applicable in microcircuits, wearable electronics and soft robotics.

[1] Group of Biomimetic Smart Materials Qingdao Institute of Bioenergy and Bioprocess Technology Chinese Academy of Sciences, Qingdao 266101, China. [2] Center of Material and Optoelectronics Engineering University of Chinese Academy of Sciences, Beijing 100049, China. [3] School of Materials Science and Engineering Harbin Institute of Technology, Harbin 150001, China. Correspondence and requests for materials should be addressed to M.L. (email: limj@qibebt.ac.cn) or to C.L. (email: Licx@qibebt.ac.cn)

Nanomaterial agglomeration and coalescence are of great importance for the formation of interconnecting conductive networks of flexible electronics applicable in soft robotics, stretchable, wearable, implantable and biomedicine devices[1,2]. Many conductive 0–1 dimensional (0–1D) nanomaterials (e.g., nanotubes, nanoparticles and nanowires of metals[3], carbon[4] and conjugated polymers[5,6]) have been endeavoured either as the colloidal inks for deposition, writing and printing, or as the fillers in organic composites[7]. When depositing and printing solid conductive colloids, capillary forces induced by solvent evaporation could assist to organise them into closely-packing layers[8]. When incorporating solid nanomaterials into organic matrix, large filler compositions were normally required to exceed the percolation threshold and form the interconnecting networks of charge transport[3,9,10], e.g., 16.2 vol% for Au nanoparticles in polyurethane[9] and 10 wt% for carbon black in polyester[10]. In most cases, agglomeration and coalescence of rigid solid nanomaterials may show contact resistance of charge transport at their physical junctions. The encapsulated layers of solid nanomaterials, which were essential for colloidal stability and compatibility with organic matrix, may also hinder their direct contact and hereby charge transport.

In contrast to rigid solid nanomaterials, droplets of liquid metal (LM, e.g., EGaIn with 75% gallium and 25% indium[8]) show the superiority in sintering into integral liquid conductors which could withstand physical deformation as diverse as bending, twisting, stretching and compression[11,12]. EGaIn droplets could be stabilised and incorporated into the printable inks and organic matrix, under the shell protection of the oxide layer (spontaneously forming and with the thickness of 0.5–3 nm[12]), ligand coordination and microgels[11,13]. Thus great potential of LM has been promised in flexible electronics because of its fluidity, metallic conductivity, negligible vapour pressure and low toxicity[12,14,15]. However, when applying in deposition, writing, printing[16], injecting into microchannels and embedding into organic matrix, EGaIn droplets were normally hindered from closely packing and spontaneously sintering due to the combination of high surface tension (624 mN m$^{-1}$) of EGaIn, protection shells and incompatibility with other materials. In particular, structural cracks frequently emerged within the layers of LM droplets due to internal stress produced by unsymmetrical capillary forces[8,11]. External interventions were usually required to rupture their encapsulating shells and release fluidic LM for droplet sintering, such as mechanical sintering[8,17] and laser sintering[18,19]. But it still remains far from being fully understood how various factors control the sintering processes of LM droplets for the formation of conductive layers on diverse substrates and interconnecting networks within organic matrix.

Recently, it was suggested that capillary forces could produce a local pressure at nanoscale up to $10^1$–$10^3$ MPa at the junctions of Ag nanowires (with the diameter of 50–90 nm), being sufficient for cold welding of the nanowires[20,21]. Herein we show that evaporation at the ambient condition (room temperature of ~20 °C, ordinary pressure of ~0.1 MPa and relative humidity (RH) of ~40%) can sinter colloidal suspensions of EGaIn droplets in the presence of biological nanofibrils (NFs, with the diameter of ~5–10 nm) as low as 0.05 wt% of cellulose, silk fibroin and amyloid, and hereafter produce conductive layers on diverse substrates and free-stranding composites with multiple functionalities, such as optical reflectivity, flexible conductivity (up to $8.9 \times 10^5$ S m$^{-1}$) and rapid responsive actuation. In contrast to conventional mechanical sintering solely on rigid substrates and laser sintering on heat-resistant substrates[8,18], this spontaneous sintering method can serve for conductive patterns, circuits and electromagnetic shield on both rigid and soft substrates (e.g., elastic, inorganic and biological). When sintering on elastic substrates, the EGaIn coating endures a tensile stain up to 200% without showing clear conductivity decay. The consolidating product may also be free-stranding with the thickness up to tens of micrometres. Consisting of a bottom EGaIn-rich layer and a top NFs-rich layer, the resultant Janus film can respond to low voltage (<3 V), photo and humidity, and show actuating behaviour with great bending speed (e.g., 120°s$^{-1}$) and repeatability, being comparable with or superior to most of soft actuators in the literature.

## Results

**Design strategies**. Biological NFs with the diameter of <10 nm are widely distributed in biomasses (Fig. 1a and Supplementary Fig. 1, 2), and function in living organisms for mechanical support (e.g., in crab shell), protection (e.g., in silk cocoon), adhesion (e.g., in biological membrane) and pathogenesis (e.g., related to Alzheimer's diseases)[22]. Technologically, silk NFs and amyloid NFs could be produced through supramolecular self-assembly (Supplementary Methods and Supplementary Fig. 1)[23,24]. Cellulose NFs (CNFs), with the average diameter of ~6 nm and aspect ratio of >$10^2$ (Fig. 1a and Supplementary Fig. 2), could be exfoliated from hardwood through a 2,2,6,6-tetramethylpiperidyl-1-oxyl (TEMPO)-mediated oxidisation[25]. With the advantages of low cost and sustainability, these NFs have seen promising applications in biomedicine, catalysis, reinforcing fillers, optoelectronics and energy-harvest[25].

Among these NFs, the synthesised CNFs typically had plentiful carboxyl groups (with the content up to 1.4 mmol g$^{-1}$) and hereby negatively charged surfaces (e.g., ζ-potential ~−50 mV at pH 7 shown in Supplementary Fig. 2, 3), being capable of forming a stable aqueous suspension (e.g., with the concentration $\phi_{CNF}$ = 0.5 wt%) at the nematic liquid-crystalline state. EGaIn, acting as a typical eutectic alloy (with a melting point ~15.8 °C), remains fluidic at room temperature with a large surface tension (624 mN m$^{-1}$) and high conductivity ($3.4 \times 10^6$ S m$^{-1}$). When sonicating (300 W and 20 kHz) EGaIn in the suspension of CNFs, an opaque grey slurry was produced with the formation of EGaIn droplets (Fig. 1b). By optimising the sonication time (e.g., 60 min in Supplementary Fig. 4) and CNFs concentration (e.g., $\phi_{CNF}$ = 0.2 wt% in Supplementary Fig. 5), the average droplet size could decrease down below 100 nm (Fig. 1c). The average droplet diameter of 50 nm was possibly achieved via further centrifugation (5000 rpm) for size grading (Supplementary Fig. 6).

To be noted, without the presence of CNFs, aqueous suspensions of EGaIn droplets were also produced with the size of >400 nm (Supplementary Fig. 5). It was reported that spontaneous oxidation assisted to encapsulate EGaIn droplets within a thin oxide shell[15]. In the presence of CNFs, EGaIn droplets were further stabilised by binding a certain amount of CNFs on their surface via crosslinking and coordination of carboxyl groups with Ga oxydate like Ga$^{3+}$ (Fig. 1c and Supplementary Fig. 7)[11]. Thus the resultant suspension had the smaller EGaIn droplets and was capable of maintaining stable up to days at pH ~7 under N$_2$ protection, with negligible precipitation and chemical oxidation (Supplementary Fig. 8, 9).

When casting the suspension of EGaIn droplets under the ambient condition, a free-standing bilayer film was obtained with the micrometric thickness. The grey top layer consisted mainly of CNFs, while the bottom layer consisted mainly of sintered EGaIn and showed smooth mirror-like surfaces (Fig. 1d, e, Supplementary Fig. 10 and Supplementary Movie 1). The thickness ratio of these two layers could be tuned by controlling the concentration ratio of CNFs and EGaIn in the suspension. Besides the free-standing films, EGaIn droplets together with CNFs could also deposit on various substrates through mask-printing,

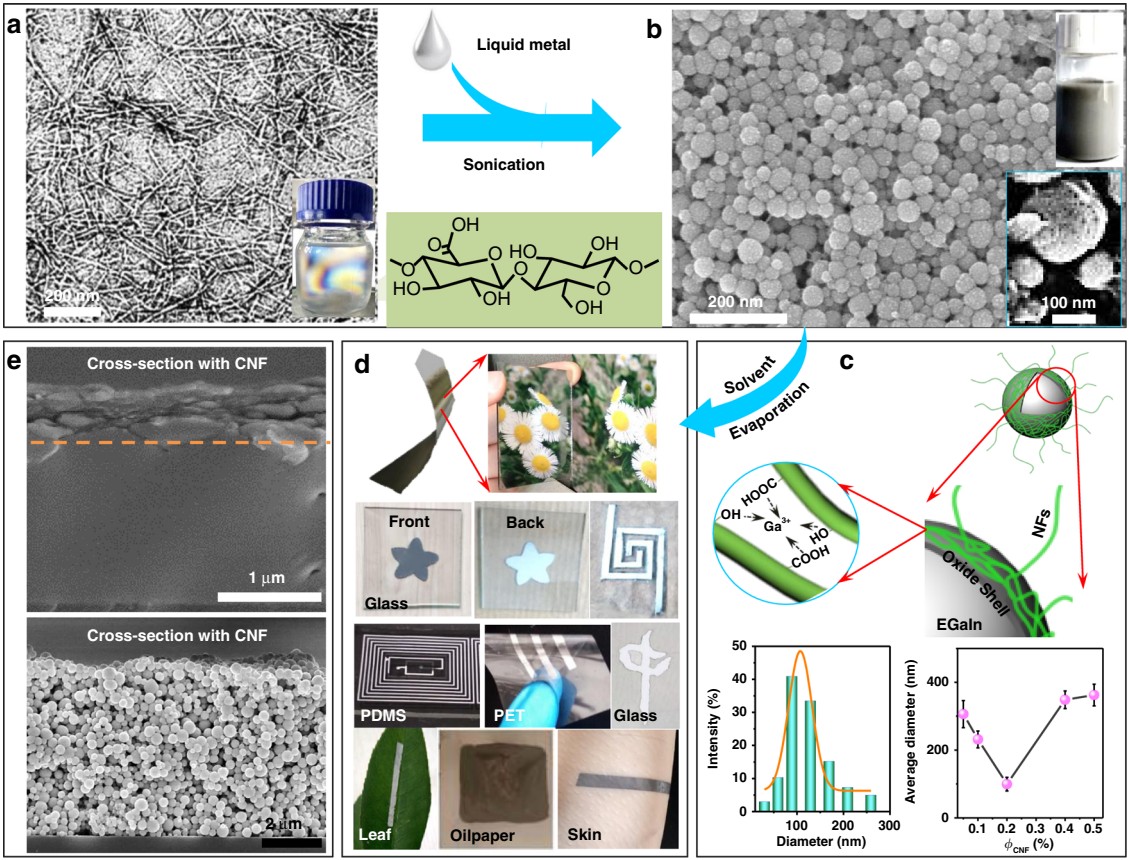

**Fig. 1** Evaporation-induced sintering of EGaIn droplets with biological NFs for free-standing films and coatings. Evaporation-induced sintering of EGaIn droplets with biological NFs. **a** Typical transmission electron microscopy (TEM) image of CNFs. The inset gives optical image of nematic CNFs suspension. **b** Typical scanning electron microscopy (SEM) image of EGaIn droplets produced by sonicating bulk EGaIn in CNFs suspension. The top-right inset gives corresponding optical image. Sonication: 60 min; Concentration ($\phi_{CNF}$): 0.2 wt%. **c** Schematic illustration of EGaIn droplet encapsulated in oxide shell and with CNFs attached on surface via interactions with $Ga^{3+}$. Diameter histogram of EGaIn droplets in **b** (Bottom left) and effect of $\phi_{CNF}$ on average EGaIn diameter after 60 min sonication (Bottom right) were given as the inset. **d** Evaporation-induced sintering into free-standing films (Top) with mirror-like bottom surface and grey top surface, and coatings (Bottom) on different substrates through mask depositing, hand-writing or drop-casting. **e** Cross-sectional SEM images of coating layers of EGaIn droplets with (Top) and without CNFs (Bottom)

channel-depositing, hand-writing and drop-casting (Fig. 1d, Supplementary Fig. 11). A bilayer coating was produced (Fig. 1d and Supplementary Fig. 12), either on the rigid substrates (e.g., glass and mica) or on the soft substrates (e.g., polyethylene terephthalate (PET), polypropylene (PP), styrene-ethylene-butylene-styrene (SEBS), polydimethylsiloxane (PDMS), leaf and skin). In contrast, without biological NFs, no free-stranding film or coating was produced either via casting or via filtration (Supplementary Fig. 13). And EGaIn droplets were deposited only randomly on the substrate without showing conspicuous coalescence and sintering (Fig. 1e, Supplementary Fig. 13). Similar results were also obtained by replacing CNFs with silk NFs or amyloid NFs (Supplementary Fig. 14).

**Flexible conductivity & degradability.** Both the free-standing films and coating on different substrates shared a common bilayer structure (Fig. 1e), suggesting the presence of a precipitation separation process during the evaporation (Supplementary Fig. 15, 16). Due to the high density (6.28 g mL$^{-1}$)[14], EGaIn droplets precipitated first and sintered into the bright layer with an electric conductivity (Supplementary Fig. 14). CNFs had relatively high colloidal stability and deposited on the EGaIn layer as the insulating layer. Inevitably, CNFs might also exist within the conductive layer because of their presence on and between the EGaIn droplets during evaporation-induced sintering

(Supplementary Fig. 17a). EGaIn droplets, when having the smaller size and attaching more CNFs, might also exist within the insulating layer because of their higher colloidal stability (Supplementary Fig. 17, 18).

The presence of CNFs not only assisted to sinter the EGaIn droplets via evaporation, but also served as the structural support for comparable mechanical properties of the free-standing Janus films (e.g., an elastic modulus up to 4.75 GPa in Supplementary Fig. 19a, b). The remaining CNFs within the EGaIn-rich layer also enabled its endurance to mild rub (see the inset of Fig. 2a), in contrast to weak surfaces when lacking CNFs (Supplementary Fig. 19c). The bottom bright layer reflected light in a way akin to bulk LM with a reflection ≥ 80% within the visible-light region (400−720 nm), whilst the top CNFs layer displayed high light absorption and thus a low reflection ≤ 20% (Fig. 2a and Supplementary Fig. 20).

When depositing on elastic substrates (e.g., SEBS), the EGaIn coating could survive hundreds of cycles of bending and twisting, without showing sensible conductivity decay (Fig. 2b and Supplementary Fig. 21). More surprisingly, this conductive layer adhered closely to the substrate during cyclic stretching, and maintained highly conductive with the strain up to 200% (Fig. 2c and Supplementary Fig. 22), in spite that the top CNFs-rich layer peeled off from the coating. In sharp contrast, the EGaIn coating without CNFs formed isolated granular morphologies after

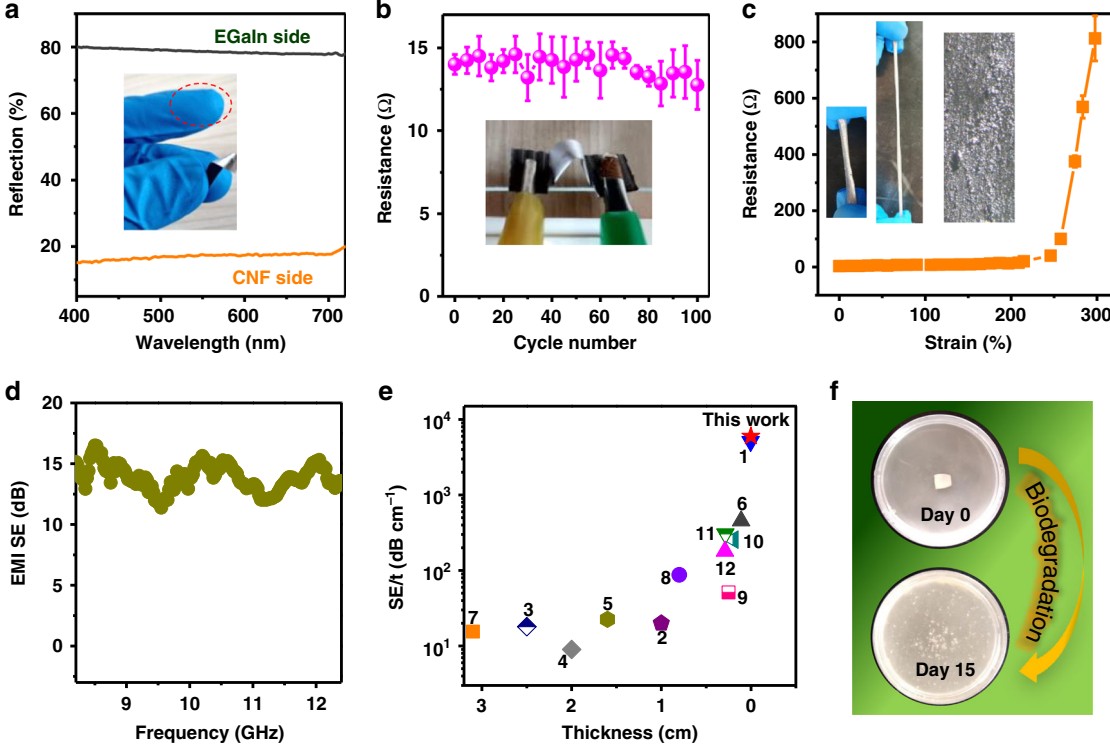

**Fig. 2** Physical characterisation of Janus EGaIn film and coating. **a** Optical reflection of CNFs-rich and EGaIn-rich sides. The inset gives endurance of EGaIn side to mild rub. **b** Cyclic bending endurance evaluated by electric resistance. EGaIn/CNFs layer thickness: 9/10 μm/μm. Bending angle 180°. **c** Stretching conductivity of EGaIn-rich layer on SEBS. EGaIn layer: 10 μm. The inset gives optical observation before (Left) and after (Middle & Right) stretching. **d** Electromagnetic shielding efficiency (EM SE) of EGaIn-rich layer after sintering. EGaIn/CNFs layer thickness: 10/10 μm/μm. **e** Comparison of $SE/t$ as function of thickness $t$. 1: Ti$_3$C$_2$T$_x$/cellulose[26]; 2: Graphene/PDMS[32]; 3: Reduced graphene oxide (rGO)[37]; 4: Carbon black/rubber[28]; 5: Carbon nanowires/graphene/PDMS[34]; 6: Carbon nanotube (CNT)/acrylonitrile-butadiene-styrene[29]; 7: Steel/PP[33]; 8: rGO/poly(3,4-ethylenedioxythiophene)[31]; 9: rGO/polyetherimide[30]; 10: Carbon/phthalonitrile[36]; 11: Ni/polyethersulfone[35]; 12: Carbon nanofibers[27]. **f** Biological degradation of free-standing film in soil extract for 0 and 15 days

stretching up to 100% (Supplementary Fig. 22). This suggested that the remaining high-aspect-ratio CNFs enabled stretchability of the conductive EGaIn-rich layer. Their polar groups also ensured the EGaIn affinity to the substrate.

Besides serving for flexible electronics, this conductive layer also exhibited an electromagnetic shielding property with a comprehensive shielding efficiency (SE) ≥ 12 dB (for the layer thickness of only ~20μm) in the X-band frequency range of 8.2–12.4 GHz (Fig. 2d). In spite of fluctuations across the tested frequency range, its overall performance is comparable or superior to many reported materials with higher thicknesses (Fig. 2e), thus being ideal for electromagnetic shielding applications[26–37]. Furthermore, the bilayer film and substrate coating could fully degrade in natural soil extract within 15 days (Fig. 2f and Supplementary Fig. 23), owing to biological and/or chemical degradation of CNFs and EGaIn. Thus this unique type of materials would offer an alternative for degradable and flexible electronics.

**Responsive actuating behaviour**. The prepared free-standing films possessed an intrinsic Janus feature, in which both the bottom and top layers showed distinct responsibility to humidity and electricity. For example, owing to different hydration abilities of CNFs and EGaIn, the Janus film delivered an unique actuating behaviour under cyclically exposing to different humidities (e.g., RH 40 and 100%), alike to reversible movement in pine cones (Supplementary Fig. 24)[38]. The actuating curvature maximised at the thickness ratio ~0.9 of the EGaIn-rich and CNFs-rich layers (see detailed calculation in Supporting Information and

Supplementary Fig. 24c). Moreover, the conductive EGaIn layer had a strong electro-thermal effect, being capable of heating under a low voltage (e.g., ≥25 °C within 3 s under 2.5 V in Fig. 3a). This Joule-heating effect would also enable an electrical actuating behaviour: A U-shaped bilayer film (24 × 2 mm² with a middle space of 0.2 mm wide; 9/10 μm/μm for EGaIn/CNFs thickness) bended up to 360° within 3 s under a constant bias of 2.0 V (RH ~ 70%), whose bending angle was tuneable by the applying voltage (Fig. 3b, Supplementary Fig. 25 and Supplementary Movie 2). When switching off the voltage, the bending angle recovered in several seconds (Fig. 3c). In analogue, this actuating behaviour was driven by cyclic dehydration and hydration of the CNFs-rich layer, in which the Joule-heating effect would dehydrate the CNFs-rich layer (Fig. 3d). This actuating behaviour showed a large bending displacement in analogue to natural flowers as well as rapidity nearly comparable to predatory motion of cabrites (Fig. 3e and Supplementary Movie 3). This actuating speed is also superior to many reported soft actuators, in the form of bilayer-structures like shape-memory and ionic polymeric composites (Fig. 3f)[39–51].

There might exist small EGaIn droplets (with the diameter <200 nm) within the CNFs-rich layer (Fig. 4a), whose photo-thermal effect had been reported in the literature[52]. This photo-thermal effect could heat the Janus film when exposing to photo radiation (Fig. 4b and Supplementary Fig. 26), and enable an actuating behaviour responsive to photo. Under a near-infrared light radiation (Wavelength of 808 nm and light density of 0.8 W cm⁻²), a Janus film (20 × 1mm²; thickness: 9/10μm/μm for EGaIn/CNFs) bended up to 90° within 2 s and recover back within <2.5 s (Fig. 4c). This

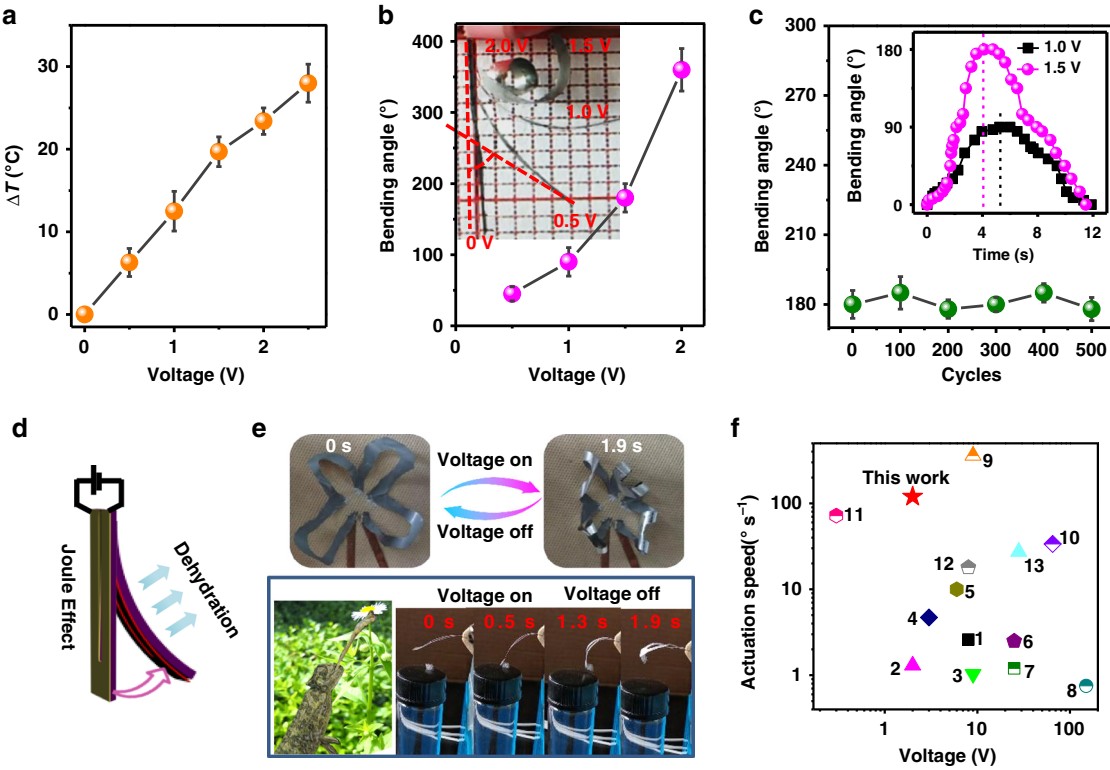

**Fig. 3** Electrically-driven actuating behaviour of Janus film. **a** Joule-heating effect at different voltages. 25 °C; Joule-heating effect (ΔT: elevated temperature from 25°C) at different voltages. Rectangular film of 24 × 2mm²; EGaIn/CNFs layer thickness: 9/10 μm/μm. **b** Bending angles at different voltages. RH: 70%. **c** Repeatable bending under at voltage of 1.5 V. The inset shows temporal variation of bending angle at constant voltage of 1.0 and 1.5 V. **d** Actuating mechanism: Dehydrating CNFs-rich layer by Joule-effect of EGaIn-rich layer. **e** Actuating behaviour in analogue to flower blossom and predatory motion of cabrites. **f** Comparison of actuating speed as function of voltage. 1: Cellulose/ionic liquid/graphene[46]; 2: Poly(styrene-alt-maleimide)/poly(vinylidene fluoride)[40]; 3: Poly(3,4-ethylenedioxythiophene)/ poly(styrene sulfonate)/ ionic liquids[47]; 4: Chitosan/CNT[39]; 5: rGO/epoxy[42]; 6: Polyurethane/CNT[45]; 7: CNT/carbon black[41]; 8: Polycyclooctene/CNT[42]; 9: Polyethylene/CNT[43]; 10: rGO/polyimide[49]; 11: MoS₂/Au;[51] 12: PP/cellulose[48]; 13: Graphene/Ag/polyimide[50]

bending-and-recovering behaviour could repeat for hundreds of times without showing obvious bending decay. Moreover, a larger bending displacement and larger velocity were produced by higher power density of NIR radiation (Fig. 4d). Besides these, this photo-thermal effect can be used to twist the bilayer film in analogue to the bean pod (Supplementary Movie 4), and drive an origami (Fig. 4e and Supplementary Movie 5). It could drive floating displacement of the Janus film on water (Fig. 4f and Supplementary Movie 6), because of locally heating surrounding water[38].

## Discussion

With the presence of biological NFs, EGaIn droplets underwent a precipitation process together with an evaporation-induced sintering process, offering an insulating NFs-rich layer over a conductive EGaIn-rich layer (with a conductivity up to ~$10^6$ S m⁻¹ with the layer thickness of >5 μm in Fig. 5a). This precipitation separation process seemed to be essential for sintering. For instance, no electric conductivity was achieved when rapidly removing the solvent through filtration and rapid drying (Supplementary Fig. 13d and 16b). The presence of CNFs was also essential for this evaporation-induced sintering process. In the suspension, the CNFs concentration as low as 0.05 wt% could assist to sinter EGaIn droplets to achieve a metallic conductivity up to $2 \times 10^5$ S m⁻¹ (Fig. 5b & Supplementary Fig. 27). And high CNFs concentrations below 0.3 wt% did not influence electric conductivity of the final EGaIn layer, probably due to their separation from the sedimental EGaIn droplets. In addition, no

evaporation-induced sintering occurred when removing free CNFs by centrifuging the EGaIn droplets (Supplementary Fig. 28).

During the sintering, CNFs unambiguously produced a local pressure large enough to rupture the encapsulating shells of EGaIn droplets, in analogue to mechanical sintering. It would produce a sharp conductivity increase (Supplementary Fig. 18). When consolidating colloidal particles, evaporation is known to give a strong capillary force as[53]:

$$F_{capillary} = 2\pi\gamma\alpha \sin\varphi \sin(\varphi + \theta) + \pi\alpha^2\sin^2\varphi\Delta P \quad (1)$$

where the first and second terms are the force arising from the surface tension and the Laplace pressure, respectively; $\varphi$ is the half-filling angle; $\gamma$ is the surface tension of the liquid; $\theta$ is the liquid–solid contact angle; $\alpha$ is the sphere radius; $\Delta P$ is the pressure difference cross the liquid surfaces, and identified as the Young-Laplace equation:

$$\Delta P = \frac{2\gamma\cos\theta}{r_p} \quad (2)$$

where $r_p$ is the capillary curvature radius. The capillary forces could also be represented as[21]:

$$F_{capillary} = 2\pi\gamma\alpha \sin\varphi \sin(\varphi + \theta) + 2\pi\gamma\alpha \cos\theta/[1 + H/2d] \quad (3)$$

where $H$ is the shortest distance between the spheres; $d$ is the immersion length of the sphere and calculated by $d = (H/2) \times [-1 + \sqrt{1 + 2V/(\pi\alpha H^2)}]$, where $V$ is the volume of liquid bridge. Colloidal agglomeration and coalescence driven by

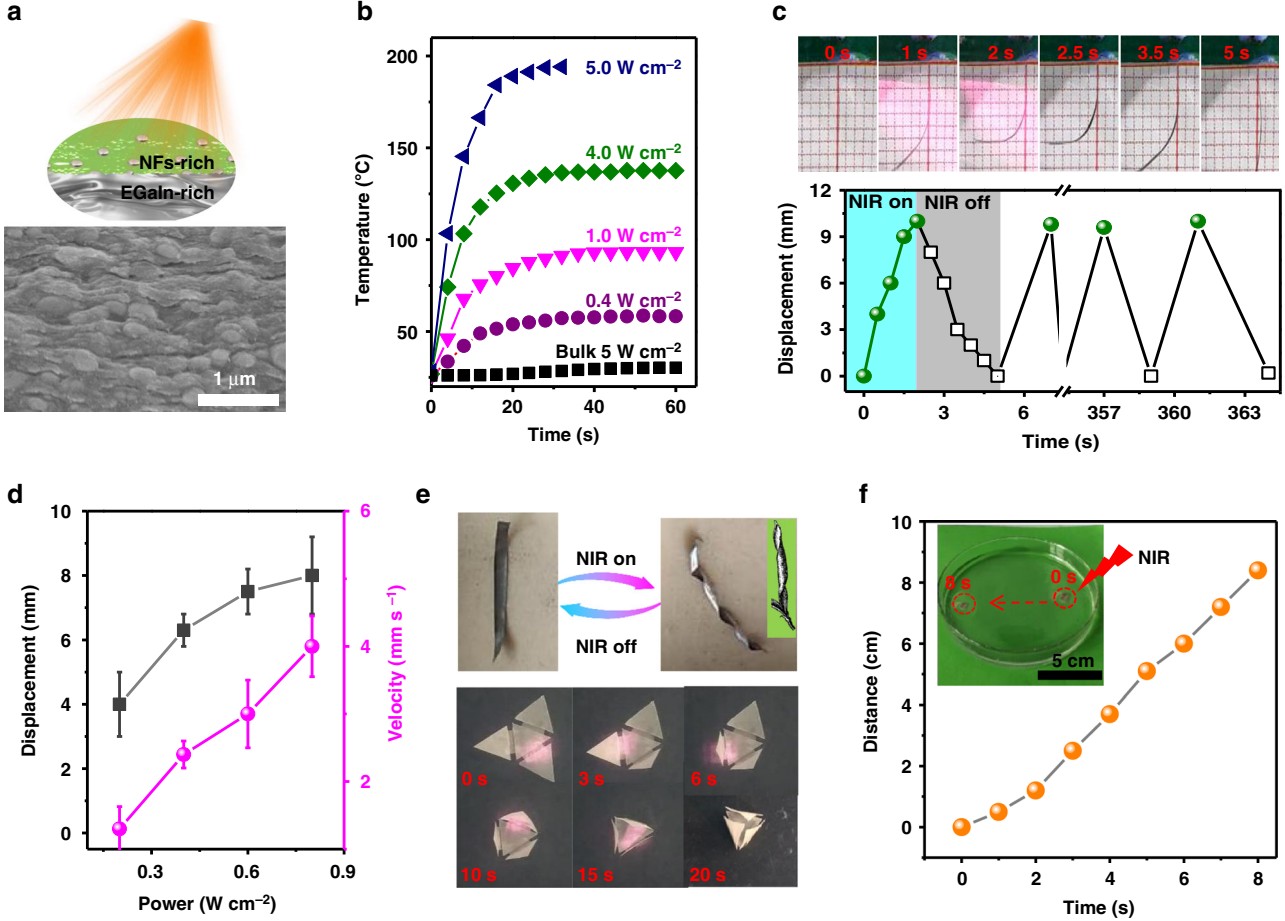

**Fig. 4** Photo-driven actuating behaviour of Janus film. **a** Schematic illustration and SEM image of EGaln droplets in CNFs-rich layer. **b** Photo-thermal effect at different NIR power densities. Rectangular film of $20 \times 10\,mm^2$; EGaln/CNFs layer thickness: 9/10 μm/μm. NIR wavelength: 808 nm. Bulk EGaln was used as the control. **c** Visual observation (Top) and repeatability (Bottom) of actuating behaviour under NIR irradiation. Power density of NIR: 0.8 W cm$^{-2}$. **d** Displacement and actuating speed at different NIR power densities. **e** Photo-driven self-twisting (Top) and origami (Bottom). The inset (top right) shows twisted *Vigna Radiata* pods. **f** Photo-driving floating displacement in water. Energy density of radiation: 0.8 W cm$^{-2}$

capillary forces widely exist in nature (e.g., biological adhesion) and colloidal technologies (e.g., inkjet printing, and self-assembly of colloidal particles)[54,55]. During evaporation of solvents, meniscus-shaped capillary bridges would form among the colloids and yield an attraction force[20,55].

Firstly, the presence of CNFs might promote $F_{capillary}$ between the EGaln droplets by decreasing their sizes and contact angles. The smaller droplet size favoured larger $\Delta P$ due to the larger capillary curvature (Fig. 5c). The attached CNFs favoured a smaller contact angle and hereby larger $\Delta P$. The increasing $\Delta P$ would contribute dominantly to $F_{capillary}$ between the EGaln droplets. For example, by assuming $\alpha = 60$ nm, $V = 10^3$ nm$^3$, $H = 5$ nm and $\theta = 5°$[56], $F_{capillary}$ would produce a local pressure up to ~13.7 MPa on EGaln droplets according to Equation 3. While only a pressure of ~0.9 MPa was obtained in absence of CNFs (detailed in Supporting Information See details in Supplementary Note 1).

Secondly, the presence of CNFs might also split the liquid bridges between the droplets. Spitting liquid bridges into small ones might lead to an increase of the capillary forces (Fig. 5d)[57]. In many biological wet adhesive systems, the adhesion forces is not only a function of contact angle, but also a nonlinear function of the number and size of liquid bridges. Multiple liquid bridges could generate the highest capillary forces without increasing the total contact area, which may exceed *van der Waals* forces by several orders of magnitude. At the same time, evaporation also

generated capillary forces among the CNFs attaching on EGaln droplets, which would produce a local pulling tension on the order of magnitude over 10 MPa[20]. Due to the large elastic modulus of CNFs up to $10^2$ GPa, this pulling tension could transmit along the CNFs and exert on the droplet shells.

Without the presence of CNFs, capillary forces among the EGaln droplets were not sufficient to rupture their oxide shells. Thus these droplets deposited on the substrate as an insulating porous layer after completely drying (Fig. 5e and Supplementary Fig. 13). When further casting a suspension (e.g., $\phi_{CNF} = 0.2$ wt%) of CNFs on the top, a portion of CNFs would infiltrate into the gaps among the EGaln droplets, and sinter them during the following evaporation (Fig. 5e). Notably, there possibly existed a thin layer of EGaln droplets between the bottom conductive layer and the top CNFs-rich layer, in which excess CNFs might form densely packing shells and prevent the droplet coalescence.

In summary, biological NFs downsized and enabled evaporation-induced sintering of LM droplets under ambient conditions into conductive free-standing functional films and coatings on substrates as diverse as rigid, soft, biological and in microfluidic channels, in sharp contrast to laser sintering on heat-resistant substrates and mechanical sintering on rigid substrates. The contribution of biological NFs was threefold: (a) Attaching on the surfaces of EGaln droplets for lower droplet size (down to 50 nm in diameter) and higher colloidal stability in the suspension; (b) During evaporation-induced sintering, biological NFs as

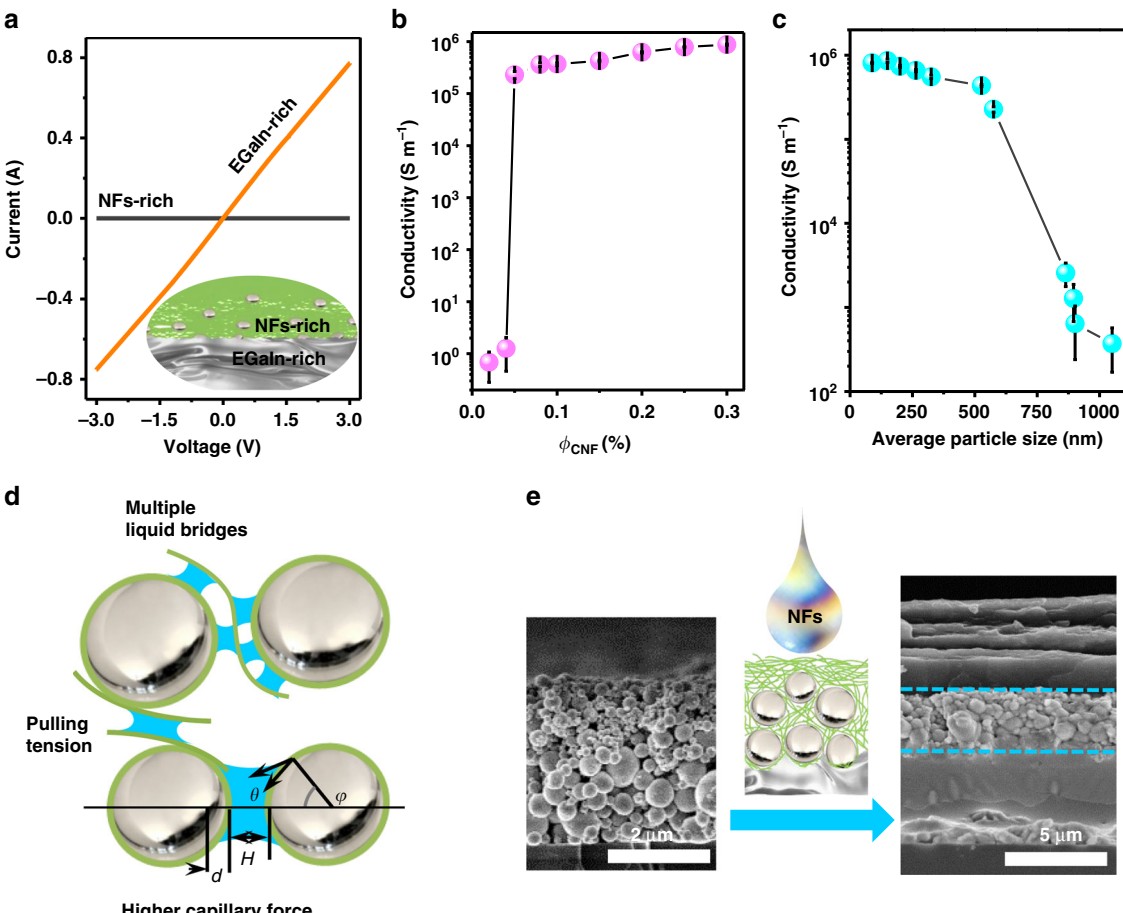

**Fig. 5** Evaporation-induced sintering mechanism of EGaIn droplets with biological NFs. **a** I–V curves of CNFs-rich and EGaIn-rich sides of free-standing films. EGaIn/CNFs layer thickness: 9/10 μm/μm. **b** Effect of CNFs concentration on evaporation-induced sintering evaluated by conductivity of EGaIn-rich side. Average EGaIn droplet size: ~120 nm. **c** Size effect of EGaIn droplets on evaporation-induced sintering evaluated by conductivity of EGaIn-rich side. **d** Schematic illustration of sintering mechanism: Capillary force promoted by biological NFs; Local pulling tension induced by biological NFs; Splitting into multiple liquid bridges. **e** Depositing NFs suspension on layer of EGaIn droplets produced without NFs for evaporation-induced sintering

low as 0.05 wt% might promote capillary forces and split liquid bridges among EGaIn droplets, and as well create high pull tension on the shells of EGaIn droplets. (c) Biological NFs had sufficient polar groups (i.e., carboxyl and hydroxyl), which also ensured adhesion and stretchability of EGaIn layers on diverse substrates by printing, casting and depositing, and thus could serve as structural matrix of EGaIn droplets for free-standing, humidity responsivity, biocompatibility and biodegradation in electronic applications. With the coexistence of NFs-rich and EGaIn-rich layers, both the resultant coating and free-standing films showed flexible conductivity (e.g., with conductivity up to $10^5$ S m$^{-1}$ at 200% strain), mirror-like reflectivity and electro-magnetic-shield, more importantly, whose actuating behaviours could response to humidity, photo and voltage. Thus this evaporation-induced approach not only extends fundamental knowledge of LM droplets and their suspensions, but also starts a new scenario of producing flexible coating and free-standing composites with flexibility, conductivity, sustainability and degradability, and applicable in microcircuits, sensing, wearable devices and soft robotics.

## Methods

**Materials**. EGaIn was provided by Shenyang Jiabei Trading Co., Ltd. (China). Needle bleached kraft pulp was bought from Hangzhou Wohua Filter Paper Co. Ltd. TEMPO was supplied by Sigma-Aldrich. Sodium hypochlorite solution (12 wt%) and NaBr were purchased from Aladdin Industrial Co. Ltd. Ultrapure water (resistivity 18.2 MΩ cm$^{-1}$) was used to prepare all aqueous solutions.

Detailed preparation of aqueous dispersions of biological NFs (including cellulose, fibroin and amyloid) was shown in Supplementary Methods.

**Preparation of EGaIn droplets**. Typically, bulk EGaIn (150 mg) was added to an aqueous solution (15 mL) of NFs. The mixture underwent sonication (BILON92-II; power of 300 W with 80% amplitude) in ice-bath water with different NFs concentrations and time periods. Size-grading was achieved by centrifuging. And free NFs in the dispersion were removed by centrifugation at 10,000 rpm.

**Evaporation-induced sintering**. The as-dispersed EGaIn droplets ($\phi_{CNF}$ 0.04–0.3 wt%, EGaIn concentration 10 mg mL$^{-1}$, and average droplet size 100–850 nm) were deposited on the substrate through drop-casting, direct hand-writing and spray-coating, following by drying at ambient conditions (~0.1 MPa, 25 ± 3 °C, RH 40 ± 5%) at least for one day for completely drying before further test. Bulk conductivity of the dried trace was measured to evaluate the sintering status of the deposited EGaIn droplets. The thickness and resistance of the dried trace were measured by SEM and digital multimeter (MS8265), respectively. For sintering on soft substrates, vacuum drying (–70 kPa) was applied. When the layer thickness of CNFs was sufficiently large, the coating on glass substrates could peeled off as free-standing films.

**Actuating test**. For voltage-responsive actuator, a Janus film was cut into a U-shape strip with a rectangle of 24 × 2 mm$^2$ and a middle space of 0.2 mm wide. Copper wires were connected to two actuator ends with Ag paste. Bending angles were recorded at the voltage of 0.5–2.0 V at RH 70%. Cyclic actuating performance was carried out at 2.0 V. For photo-responsive actuator, a Janus film (20 × 1 mm$^2$) was equilibrated at 25 °C and RH ~70% for 24 h. A NIR laser with tunable power (wavelength of 808 nm, maximum output power of 1.0 W, Xi'an Minghui Optoelectronic Technology Co., Ltd.) was used as the NIR source.

**Stretchable conductivity test**. A sintered coating (with 9/20 μm/μm in thickness of CNFs and EGaIn) was prepared on a strip of SEBS ($20 \times 5 \times 0.5$ mm$^3$). The stretching test was performed on a mechanical tester at a speed of 2 mm min$^{-1}$.

**Electromagnetic shielding test**. The performance was measured on an Agilent N5227A vector network analyser using an APC-7 connector as the coaxial test cell in the X-band frequency range of 8.2–12.4 GHz. The background from the substrate was deducted according to a simple addition rule.

**Characterisation**. Field emission scanning electron microscopy (FESEM, Hitachi S-4800, Japan) with X-ray energy dispersive spectrometry (EDS) was used to characterise microstructure and elemental mapping at an acceleration voltage of 10 kV. TEM (Hitachi H-7650) measurements were performed at a voltage of 100 kV. Tensile experiments were performed on Electromechanical Universal Testing Machine (CMT6503, MTS Industrial Systems Co., Ltd. China). Photo reflection spectra were recorded in the visible light region (400−720 nm) with a Hitachi U-4100 spectrophotometer (Japan).

## Data availability

Data supporting the findings of this work are available within the paper and its Supplementary Information files and from the corresponding authors upon reasonable request.

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

## Acknowledgements

National Natural Science Foundation of China (Nos. 21474125 and 51608509), Chinese "1000 Youth Talent Program", Shandong "Taishan Youth Scholar Program", Shandong Provincial Natural Science Foundation (Nos. JQ201609 and ZR2016EEB25), "135" Projects Fund of CAS QIBEBT Director Innovation Foundation and Shandong Collaborative Innovation Centre for Marine Biomass Fiber Materials and Textiles are kindly acknowledged for financial support. The authors also thank Dr. Anle Ge in Qingdao Institute of Bioenergy and Bioprocess Technology, Chinese Academy of Sciences for the contribution to preparation of microfluidic channels and Prof. Shu Lin in Harbin Institute of Technology for the contribution to analysis of electromagnetic interference shielding results.

## Author contributions

X.L., M. L, J.X., J.Y. and Z.Y. performed the experiments. C.L. and M.L performed the analysis and wrote the paper. C.L. directed this study.

## Additional information

**Competing interests:** The authors declare no competing interests.

