## [Peer Review File · Nature Communications]

Reviewers' comments:

Reviewer #1 (Remarks to the Author):

The manuscript entitled "Evaporatively sintering liquid metal droplets with biological nanofibrils for flexible conductivity and responsive actuation" by Chaoxu Li et al. describes a novel method for sintering of liquid metal droplets enabling a wide range of exciting applications presented in the manuscript.

I would recommend the work to be accepted following a minor revision, addressing my following comments:

1. Please improve the quality of Figures 2b and 2c, as the circles used for presenting the points is larger than the error bars.
2. Figure 2b: please explain the sharp increase of conductivity at $\phi=0.05$.
3. Figure 3d: please describe the relatively large fluctuations of EMI SE across the frequency range.
4. Figure 3f: please provide suggestions for accelerating the biodegradation of free standing films.
5. Please comment on the effect of oxidation on the performance of liquid metal layer. In other words, please explain the long-term performance of such devices.

Reviewer #2 (Remarks to the Author):

The authors propose a technique to sinter nanoscale droplets of liquid metal (LM) using an evaporation-controlled technique. This is accomplished by coating the LM droplets with organic nanofibers (e.g cellulose) and then allowing capillary forces to apply pressure to the droplets and cause them to rupture and coalesce. The authors perform electrical and electromechanical characterization of LM films produced using this technique. They also present results for bending actuation with these films through a combination of solvent evaporation and Joule heating or photo-thermal interaction.

This study presents an intriguing solution to an important challenge in liquid metal electronics. However, although the mechanism for evaporation-controlled LM sintering is clearly described, I'm having a difficult time following the actual experimental evidence for this. While the authors do an extensive job of characterizing the sintered films after it is made, they present virtually no characterization of the actual sintering process. Given that the evaporation-controlled sintering method is the central claim and novelty of this work, it demands more complete experimental validation. At the very least, I'd expect to see the following:

- 1) Conductivity as a function of time following deposition of the film on the substrate. The time rate of increase in conductivity will be indicative of the nature and timescales associated with the sintering process.
- 2) Visual evidence of change in suspension morphology over time. This would likely have to be accomplished by taking a recording with an environmental SEM. As the solvent evaporates, it should be visually apparent that the droplets are rupturing and coalescing.
- 3) TEM with direct visual evidence of cellulose nanofibers embedded in Ga₂O₃ shell, as illustrated in Fig. 1C.

In addition, there are some minor issues in terms of presentation and referencing:

- 4) Given the focus on evaporation-controlled sintering, there is excessive attention to thermal actuation with the Janus effect. It needs to be more clear how this additional work is tied to the main claims of the study.

5) Fig. 1 is cluttered with a lot of images and illustrations that don't appear to be essential (Fig. 1a and 1b especially). The figure needs to be cleaned up.

6) Fig. 2e needs labels to make it more clear what is being depicted.

7) The choice of references is confusing and incomplete. For example, Ref. 2 isn't related to nanomaterial agglomeration for conductive networks. Also, the authors are missing several relevant articles on LM droplet sintering, including (but not limited to) the following [this Reviewer was not involved in any of these studies]:

Lin, Y., Cooper, C., Wang, M., Adams, J.J., Genzer, J. and Dickey, M.D., 2015. Handwritten, soft circuit boards and antennas using liquid metal nanoparticles. *Small*, 11(48), pp.6397-6403.

Wang, J., Cai, G., Li, S., Gao, D., Xiong, J. and Lee, P.S., 2018. Printable Superelastic Conductors with Extreme Stretchability and Robust Cycling Endurance Enabled by Liquid-Metal Particles. *Advanced Materials*, 30(16), p.1706157.

Reviewer #3 (Remarks to the Author):

The authors report an effect of "superiority in coalescing" of EGaIn droplets in presence of CNFs' nanofibers. The obtained double sided free stand films possess relatively high mechanical properties, as well as high reflectivity, and actuating behavior responsive to humidity, voltage and photo.

However, in my opinion the manuscript is written in a strange manner, i.e. it looks like more "advertising presentation" than the original scientific paper. I am looking to the root of the effect and want to understand its physics.

The paper requires a major revision based on the comments shown below:

Title:

Better: Evaporative sintering?!

And may be better fibers, while fibrils may be also OK

(a) Introduction is too long. The problem can be explained and formulated in two paragraphs (paragraphs 1 and 2 can be easily omitted, the rest shortened. Try to be very specific.

However, the following specific issues should be addressed:

- For the general reader please explain the specific properties of the eutectic GaIn (EGaIn) alloy;
- Please describe the process of "natural evaporation", as you understand it
- Please explain why the use of biological fibers is so important. For sure, the general statement on "a local pressure at nanoscale" does not directly relate to the nature of the fibers.

(b) Results. Figure 1, in the present form, may be moved to the supplementary file. Please do not use cartoons in presenting your data.

In Figure 1a please show the typical TEM image of the fibers, which was used to obtain their diameter distribution and the corresponding distribution;

Figure 1b please show the typical SEM/TEM image that was used to obtain the droplets distribution before sonication and corresponding droplets size distribution; Figure 1c: if you can, show the image of the suspension;

Figure 1d: show the nice image of the cross section for the obtained film.

Figure 1f: show the nice image of the cross section for the film formed without use of CNFs.

Figure 2: only conductivity data should be presented.

(c) Mechanism. Should be moved to the discussion part.

This is the important part of the paper. First, here you should explain in more details what does it mean "suspension evaporation" and "evaporative sintering".

Next:

- It is understandable that “Due to the high density (6.28 g mL^{-1}), 16 EGaIn droplets precipitated first ... (with the layer thickness of $>5 \mu\text{m}$). CNFs had relatively high colloidal stability and deposited on the EGaIn layer as the insulating layer.”
 - However the following important statement is not well documented “CNFs might also exist within the conductive layer because of their presence on and between the EGaIn droplets during evaporative sintering. EGaIn droplets, when having the smaller size and attaching more CNFs, might also exist within the insulating layer because of their higher colloidal stability.”
- Can you provide some evidences for the above: e.g. based on TEM studies of the film?

More important: Equation 3 does not contain directly any characteristics of the CNFs and calculated value of $FCapillary$ could be present for the droplets themselves. It can be speculated that CNTs influence the wetting angle, but in this case, the wetting studies in droplets-CNf system are critical. Bottom line, the Eq. (3) without additional discussion does not explain the observed effect.

Next, re “spitting the liquid bridges”. The claim is “...a portion of CNFs would infiltrate into the gaps among the EGaIn droplets, and sinter them during the following evaporation...” It is completely speculation. Can you provide any evidences for this mechanism? (see above comments to the more detailed characterization of the obtained films). Here you may use some nice “cartoon”-type figure to show all mentioned parameters and processes.

(d) Properties. Please explain what is so special in that “The bottom bright layer reflected light in a way akin to bulk LM with a reflection $\geq 80\%$ within the visible-light region (400–720 nm)”.

Responses to comments of the reviewers

Responses to Reviewer #1:

The manuscript entitled "Evaporatively sintering liquid metal droplets with biological nanofibrils for flexible conductivity and responsive actuation" by Chaoxu Li et al. describes a novel method for sintering of liquid metal droplets enabling a wide range of exciting applications presented in the manuscript.

I would recommend the work to be accepted following a minor revision, addressing my following comments:

Reply 1: We would like to thank the reviewer for carefully and kindly commenting the manuscript. All comments have been carefully considered. The corresponding responses are highlighted (blue for relocation and red for modification) in the manuscript and listed in detail as the following.

1. Please improve the quality of Figures 2b and 2c, as the circles used for presenting the points is larger than the error bars.

Reply 1-1: Thanks very much for this valuable suggestion!

The quality of Fig. 2b & c (revised as Fig. 5b & c) was improved by reducing the circle sizes and changing their colors as shown in Fig. R1-1a & b. So the error bars were presented much clearly.

For further improving the visibility of data points with large numerical values, zoom-in images of Fig. 2b & c (revised as Fig. 5b & c) with linear coordinate were also given as shown in Fig. R1-1c & d. And corresponding revisions have been made in Fig. 5b & c and Supplementary Figure 27 in the revised manuscript.

Fig. R1-1. (a & c) Effects of CNFs concentration on evaporation-induced sintering evaluated by conductivity of EGaIn rich side. Average EGaIn droplet size: ~ 120 nm. (b & d) Size effect of EGaIn droplets on evaporation-induced sintering evaluated by conductivity of EGaIn-rich side. Fig. c & d are zoom-in images of Fig. a & b, respectively.

2. Figure 2b: please explain the sharp increase of conductivity at $\phi = 0.05$.

Reply 1-2: That is a good question!

To figure out the underlying mechanism of sharp conductivity increase at $\phi_{CNF} = 0.05$ wt%, SEM images of stacked EGaIn-CNFs composites were observed as shown in Fig. R1-2.

Evaporating dispersions of EGaIn droplets with a low CNFs concentration (e.g., $\phi_{CNF} \leq 0.03$ wt%) only led to a stack of monodispersed EGaIn droplets (Fig. R1-2a) with an ultralow conductivity less than $0.7 S m^{-1}$ due to the presence of an insulative oxide shell around each EGaIn droplet. With increasing ϕ_{CNF} up to 0.04 wt%, capillary force during evaporation was increased high enough to rupture some EGaIn droplets as shown in Fig. R1-2b. However, the sintered EGaIn droplets did not connect together within the whole EGaIn-CNFs composite film (Fig. R1-2b). Therefore, the

conductivity of the film was still very low as $\sim 0.7 \text{ S m}^{-1}$. At $\phi_{\text{CNF}} = 0.05 \text{ wt\%}$, capillary force was high enough to rupture and sinter most EGaIn droplets to form a conductive path within the film (Fig. R1-2c). So a metallic conductivity up to 10^5 S m^{-1} was obtained.

Fig. R1-2. Cross-section (Top) and back-surface (Bottom) SEM images of the stacked EGaIn-CNFs layers in the presence of CNFs with different concentrations.

And the corresponding revision was added in the revised manuscript as follows:

In Page 10, “In the suspension, the CNF concentration as low as 0.05 wt% could assist to sinter EGaIn droplets (Fig. 2b).” has been changed to “In the suspension, the CNFs concentration as low as 0.05 wt% could assist to sinter EGaIn droplets to achieve a metallic conductivity up to $2 \times 10^5 \text{ S m}^{-1}$ (Fig. 5b & Supplementary Figure 27).”.

Fig. R1-2 was also added in the revised manuscript as Supplementary Figure 27 with caption of “(a & b) Effects of CNF concentration (a) and droplet size (b) on evaporation-induced sintering evaluated by conductivity of EGaIn-rich side. EGaIn droplet size: $\sim 120 \text{ nm}$. (c-e) Cross-section (Top) and back-surface (Bottom) SEM images of EGaIn layers producing with different CNF concentrations. ϕ_{CNF} : 0.03 wt% (c), 0.04 wt% (d), 0.05 wt% (e). With a low CNF concentration (e.g., $\phi_{\text{CNF}} \leq 0.03 \text{ wt\%}$), no trace of sintering was detected, and thus resulted in an ultralow conductivity due to the presence of insulating oxide shells of EGaIn droplets. At $\phi_{\text{CNF}} > 0.04 \text{ wt\%}$, EGaIn droplets started to be sintered and produce a conductivity increase. At $\phi_{\text{CNF}} = 0.05 \text{ wt\%}$, EGaIn droplets were sintered and form a relatively homogenous conductive layer with a metallic conductivity up to 10^5 S m^{-1} ”.

3. Figure 3d: please describe the relatively large fluctuations of EMI SE across the frequency range.

Reply 1-3: We appreciate this comment!

The EMI shielding effect of film materials should mainly lie in absorptions and reflections. Compared with conventional electromagnetic composites with nanofillers, reflection may contribute largely to the EMI shielding of EGaIn film owing to its smooth surface (i.e., the surface attached to the glass substrate). These reflection waves, including multiple reflections by the EGaIn film and waveguide wall as shown in Fig. R1-3, easily cause interference and stationary waves with the incident electromagnetic waves in the chamber of incidence system. The incident electromagnetic waves arriving at the EGaIn film should be an overlay of multiple waves with many uncertainties (e.g., frequency and power), which may induce fluctuations of EMI SE value obtained from the end of test system. Similar fluctuations were reported in the previous studies. (*ACS Appl. Mater. Interfaces* 2016, 8, 33230; *Adv. Funct. Mater.* 2016, 26, 447). But the overall EMI SE of bilayer film was comparable to the reported materials even with a higher thickness (Fig. 2e).

Fig. R1-3. EMI SE test system.

And the corresponding revision was added in the revised manuscript as follows:

In Page 8, “being comparable to many reported materials at the lower thickness (Figure. 3e) and ideal for electromagnetic shielding applications.” has been changed to “In spite of fluctuations across the tested frequency range, its overall performance is comparable or superior to many

reported materials with the higher thicknesses (Fig. 2e) and ideal for electromagnetic shielding applications.”.

4. Figure 3f: please provide suggestions for accelerating the biodegradation of free standing films.

Reply 1-4: This is a good question!

The biodegradation of films containing CNFs and EGAIn was carried out in the natural soil extract at 20 °C. Based on the previous studies (e.g., Carbohydrate Polymers, 2015, 132, 1-8.), the biodecomposition should be governed by various microorganisms such as bacteria and fungi. Therefore, surrounding factors that facilitate the growth and reproduction of these various microorganisms will accelerate the biodegradation of the film.

According to the reviewer’s specific suggestion, the biodegradation test was carried out in the natural soil extract again yet at an elevated temperature of 30 °C. As shown in Fig. R1-4, the bilayer film could fully degrade within 15 days about one time faster than that at 20 °C. Moreover, pre-cultivation of the nature soil extract before the film addition could also accelerate its biodegradation. For example, we performed the film biodegradation test after cultivation of the nature soil extract with 5 mg mL⁻¹ glucose for one day, only 10 days was needed for full biodegradation (Fig. R1-4).

Fig. R1-4. Biodegradation of Janus film for indicated periods of time in soil extract before (a) and after (b) pre-cultivation with 5 mg mL⁻¹ glucose for one day at 30 °C.

And the corresponding revision was added in the revised manuscript as follows:

In Page 8, “Furthermore, the bilayer film and substrate coating could fully degrade in natural soil extract **within 30 days (Fig. 3f and Supplementary Figure 22)**, owing to biological and/or chemical degradation of CNFs and EGaIn..” has been changed to “Furthermore, the bilayer film and substrate coating could fully degrade in natural soil extract **within 15 days (Fig. 2f and Supplementary Figure 23)**, owing to biological and/or chemical degradation of CNFs and EGaIn.”.

Fig. R1-4 was also added in the revised manuscript as Supplementary Figure 23 with caption of “(a-b) Biological degradation of EGaIn Janus film for indicated periods of time in soil extract with (a) and without (b) pre-cultivation in 5 mg mL^{-1} glucose for 24 h at $30 \text{ }^\circ\text{C}$. (c) Chemical degradation of EGaIn Janus film for indicated periods in 1 M NaOH ”.

5. Please comment on the effect of oxidation on the performance of liquid metal layer. In other words, please explain the long-term performance of such devices.

Reply 1-5: We appreciate this comment very much.

In this work, the Janus film could serve as U-shaped actuator with long-term cyclic stability, e.g., 500 cycles without obvious decay in bending performance and conductivity (Fig. R1-5a & b).

To investigate the effect of surface oxidation of the liquid metal layer, SEM was carried out to observe the liquid metal surface after different cycle numbers (Fig. R1-5c).

For fresh peeled film, its surface is smooth. Because of oxygen in air a thin oxide layer (mainly Ga_2O_3) will immediately form on the surface of liquid metal (ACS Nano 2017, 11, 7440; Adv. Funct. Mater. 2018, 28, 1804197). The oxidation will also happen when actuator bending with oxide layer broken and new liquid metal exposed, and the thin and flexible oxide layer will be wrinkled when actuator unbending. After bending-unbending cycles as actuator, the number of wrinkles on the surface of liquid metal layer becomes more and turns stable after about 400th cycles' actuation. That should be due to the fact that the wrinkled oxide layer could unfold when actuator bending and fully cover the unoxidized liquid metal, protecting the liquid metal from further oxidation as a self-passivated skin. Therefore, in spite of surface oxidation the passivation effect of the wrinkled oxide layer could retain the conductivity and actuation performance in long-term cycles.

Fig. R1-5. (a) Repeatability of bending under at voltage of 1.5 V. The inset shows the corresponding visual observation. (b) Conductivity of the device after different repeatable bending cycles. (c) Surface SEM images of the liquid metal layer after different repeatable bending cycles.

Responses to Reviewer #2:

The authors propose a technique to sinter nanoscale droplets of liquid metal (LM) using an evaporation-controlled technique. This is accomplished by coating the LM droplets with organic nanofibers (e.g cellulose) and then allowing capillary forces to apply pressure to the droplets and cause them to rupture and coalesce. The authors perform electrical and electromechanical characterization of LM films produced using this technique. They also present results for bending actuation with these films through a combination of solvent evaporation and Joule heating or photo-thermal interaction.

This study presents an intriguing solution to an important challenge in liquid metal electronics. However, although the mechanism for evaporation-controlled LM sintering is clearly described, I'm having a difficult time following the actual experimental evidence for this. While the authors do an extensive job of characterizing the sintered films after it is made, they present virtually no characterization of the actual sintering process. Given that the evaporation-controlled sintering method is the central claim and novelty of this work, it demands more complete experimental validation. At the very least, I'd expect to see the following:

***Reply 2:** Thank you very much for your pertinent comments and valuable suggestions. We hope that our answers below and the changes highlighted (blue for relocation and red for modification) in the manuscript could fully address the reviewer's concerns.*

1) Conductivity as a function of time following deposition of the film on the substrate. The time rate of increase in conductivity will be indicative of the nature and timescales associated with the sintering process.

***Reply 2-1:** We appreciate this suggestion!*

Accordingly, conductivity as a function of time during the evaporation-induced sintering process was carried out as shown in Fig R2-1.

At the beginning, the conductivity was very low which should be the ionic conductivity of the dispersion. With the solvent evaporation going on, the water became less enough and meniscus would emerge within the EGaIn droplets-CNFs stacks. And then capillary force was high enough to rupture and sinter some EGaIn droplets to form a conductive path within the film. So a sharp

conductivity increase was obtained. Afterwards, a modest increase in conductivity was found because more EGaIn droplets were sintered and more conductive path formed, then the conductivity became stable indicating the full sintering and thorough drying (Fig R2-1 & Supplementary Figure 18a).

Fig. R2-1. Conductivity of the deposited film on the glass versus time. Volume of 0.5 mL liquid metal droplets dispersion ($\phi_{CNF} \leq 0.2$ wt%) in a $1.5 \times 1 \times 1$ cm³ glass chamber.

And corresponding revisions have been made as follows:

In Page 10, “When consolidating colloidal particles, evaporation is known to give a capillary force as:” has been changed to “*It would produce a sharp conductivity increase (Supplementary Figure 18).* When consolidating colloidal particles, evaporation is known to give a strong capillary force as”.

And the Fig. R2-1 has been added in the Supplementary Figure 18a in the revised manuscript together with the caption of “(a) *Conductivity variation versus time during drying suspension of EGaIn droplets on glass. Suspension volume of EGaIn droplets: 0.5 mL; ϕ_{CNF} : ≤ 0.2 wt%; Glass substrate: 1.5×1 cm². Before solvent evaporation, only a low ionic conductivity was detected in the fluid layer of EGaIn droplets. During solvent evaporation, the capillary force may be high enough to sinter EGaIn droplets and produce a conductivity increase.*”.

2) Visual evidence of change in suspension morphology over time. This would likely have to be accomplished by taking a recording with an environmental SEM. As the solvent evaporates, it should be visually apparent that the droplets are rupturing and coalescing.

Reply 2-2: Thanks very much for this valuable suggestion!

Cross-sectional SEM images at different time periods during solvent evaporation were observed. Before test, the specimen was frozen in liquid nitrogen ($-196\text{ }^{\circ}\text{C}$) for 30 min and dried in a freeze-drier (FD-1C-50, Beijing BoYiKang) at $-50\text{ }^{\circ}\text{C}$. As shown in Fig. R2-2, the rupturing and coalescing of the EGaIn droplets were visually apparent as the solvent evaporated.

Fig. R2-2. (a) Cross-sectional SEM images of the film at different time periods during solvent evaporation. (b) Zoom-in images of Fig. R2-2a for the bottom part of the film at the last one hour during evaporation.

And the obtained SEM images have been added into the Supplementary Figure 18b-c of revised manuscript with the caption:“(b) Cross-sectional SEM image when drying suspension of EGaIn droplets on glass for indicated periods of time. Each specimen underwent a freeze-drying process before SEM characterization. (c) SEM images for characterization of sintering process of EGaIn droplets during solvent evaporation. Before solvent evaporation, EGaIn droplets was apt to precipitate first on the bottom due to its large density of $\sim 6\text{ g cm}^{-3}$, while free CNFs which were not attached on EGaIn droplets tended to keep colloiddally stable in the suspension (Supplementary Figure 3, 8 and 9). At the final stage of solvent evaporation, capillary force was high enough to sinter EGaIn droplets and formed a conductive EGaIn-rich layer. Part of CNFs would stay over the EGaIn-rich layer and dry into a CNFs-rich layer. To be noted, part of EGaIn droplets may had the

smaller size and attach a large amount of CNFs, which would show higher colloidal stability and thus stay in the CNFs-rich layer after completely drying.”

3) TEM with direct visual evidence of cellulose nanofibers embedded in Ga₂O₃ shell, as illustrated in Fig. 1C.

Reply 2-3: *We sincerely appreciate your comment and the valuable suggestion.*

In this study, CNFs should be bound around the EGaIn droplet and its oxide shell. It is well known that upon sonicating EGaIn in pure water fresh exposed EGaIn will be oxidized immediately to form a thin oxide shell (ACS Nano 2017, 11, 7440), and at the same time a little Ga³⁺ will be released from the surface of EGaIn droplets (as proved in our previous work of Adv. Funct. Mater. 2018, 28, 1804197). When adding CNFs, a certain amount of CNFs will bind on the surface of EGaIn droplets via coordination of carboxyl groups with Ga oxydate and be cross-linked around the droplets by released Ga³⁺ as shown in Fig. R2-3.

Fig. R2-3. Typical SEM (a) and TEM image (b) of EGaIn droplets encapsulated in oxide shell and wrapped with a certain amount of CNFs on their surface.

The authors are sorry for the poor illustration in Fig. 1c, and corresponding revisions have been made as follows:

*In Page 6, “In the presence of CNFs, EGaIn droplets **will be were** further stabilized by binding a certain amount of CNFs **via coordination of carboxyl groups with Ga oxide (Fig. 1c and Supplementary Figure 7).**” has been changed to “In the presence of CNFs, EGaIn droplets **were** further stabilized by binding a certain amount of CNFs **on their surface via crosslinking and coordination of carboxyl groups with Ga oxydate like Ga³⁺ (Fig. 1c and Supplementary Figure 7).**”.*

In Page 20, the caption of Fig. 1c has been revised to “*Schematic structure of EGaIn droplet encapsulated in oxide shell and with CNFs attached on its surface via interactions with Ga³⁺.*”.

In addition, there are some minor issues in terms of presentation and referencing:

4) Given the focus on evaporation-controlled sintering, there is excessive attention to thermal actuation with the Janus effect. It needs to be more clear how this additional work is tied to the main claims of the study.

Reply 2-4: *We sincerely appreciate this comment and the valuable suggestion!*

*In this work, the presence of CNFs not only assisted to sinter the EGaIn droplets via the suspension evaporation, but also served as the structural stabilizer for EGaIn-CNFs composites (e.g., coatings and free-standing Janus films). The resultant EGaIn-CNFs composites possess a unique structure with an insulating CNFs-rich layer and a conductive EGaIn-rich layer. Notably, the two layers were tightly coupled together, the binding force of which was much higher than that of bilayer liquid metal-organic structures produced from other methods. And there exists small EGaIn droplets (with the diameter <200 nm) within the CNFs-rich layer, who have unique photo-thermal effect. Therefore, the authors think one task of this work is to present the phenomenon of evaporation-induced sintering and as well to discuss the possible underlying mechanism (subsections of “Design strategies” and “Mechanism analysis”). And the other task is to fully reveal the functions and potential applications of the composites (subsections of “Flexible conductivity & degradability” and “Responsive actuating behavior”) for the audience to exhibit the significance of the method of evaporation-induced sintering. We surprisingly found that the resultant Janus film exhibited superior actuation properties as they could respond to low voltage (<3 V), photo and humidity, and show actuating behavior with great bending speed (e.g., 120° s⁻¹) and repeatability, being superior to most of soft actuators in the literatures. So the authors added these results in the main text of the manuscript, and we believe these results are interesting and can attract a wide audience of the journal **Nature Communications**. And an explicit statement for the close association between nanomaterial coalescence and flexible electronics (e.g., soft robotics and wearable devices) has been made in the beginning of the Introduction section.*

5) Fig. 1 is cluttered with a lot of images and illustrations that don't appear to be essential (Fig. 1a and 1b especially). The figure needs to be cleaned up.

Reply 2-5: We appreciate this suggestion very much!

Fig. 1 has been revised carefully according to this specific comment as shown in Fig. R2-4, and other images and cartoons have been removed into the Supplementary Information (Supplementary Figure 1) as shown in Fig. R2-5:

Fig. R2-4. **Evaporation-induced sintering of EGaIn droplets with biological NFs for free-standing films and coatings.** (a) Typical TEM image of CNFs. The inset gives optical image of nematic CNFs suspension. (b) Typical SEM image of EGaIn droplets produced by sonicating bulk EGaIn in CNF suspension. The top-right inset gives corresponding optical image. Sonication: 60 min; Concentration (ϕ_{CNF}): 0.2 wt%. (c) Schematic illustration of EGaIn droplet encapsulated in oxide shell and with CNFs attached on surface via interactions with Ga^{3+} . Diameter histogram of EGaIn droplets in Fig. 1b (Bottom left) and effect of ϕ_{CNF} on average EGaIn diameter after 60 min

sonication (Bottom right) were given as the inset. (d) Evaporation-induced sintering into free-standing films (Top) with mirror-like bottom surface and grey top surface, and coatings (Bottom) on different substrates through mask depositing, hand-writing or drop-casting. (e) Cross-sectional SEM images of coating layers of EGaIn droplets with (Top) and without CNFs (Bottom).

Fig. R2-5. (a) Schematic illustration of synthesizing biological NFs from biomass via liquid exfoliation. TEM images and diameter histogram of amyloid NFs (b & c) and silk NFs (d & e). The inset gives visual observation of corresponding aqueous suspensions (0.1 wt%).

6) Fig. 2e needs labels to make it more clear what is being depicted.

Reply 2-6: We are sorry for losing this important information!

Scale label has been added into the corresponding figures as follows:

Figure R2-6. Depositing NFs suspension on layer of EGaIn droplets produced without NFs for evaporation-induced sintering.

7) The choice of references is confusing and incomplete. For example, Ref. 2 isn't related to nanomaterial agglomeration for conductive networks. Also, the authors are missing several relevant articles on LM droplet sintering, including (but not limited to) the following [this Reviewer was not involved in any of these studies]:

Lin, Y., Cooper, C., Wang, M., Adams, J.J., Genzer, J. and Dickey, M.D., 2015. Handwritten, soft circuit boards and antennas using liquid metal nanoparticles. *Small*, 11(48), pp.6397-6403.

Wang, J., Cai, G., Li, S., Gao, D., Xiong, J. and Lee, P.S., 2018. Printable Superelastic Conductors with Extreme Stretchability and Robust Cycling Endurance Enabled by Liquid - Metal Particles. *Advanced Materials*, 30(16), p.1706157.

Reply 2-7: *We appreciate this comment very much!*

Ref. 2 has been changed to a more appropriate reference. And we are sorry for missing some very important relevant references.

*According to the specific comment, we have cited these significant papers in the revised manuscript as Ref. 16-17, which greatly improves the quality of our manuscript to merit its publication in **Nature Communications**. And the revisions have been made in the revised manuscript as follows:*

*In page 16, “2. Zhu, B., Gong, S. & Cheng, W. Softening gold for elastronics. *Chem. Soc. Rev.* 48, 1668-1711 (2019).”*

*In page 17, “16. Wang, J., Cai, G., Li, S., Gao, D., Xiong, J. & Lee, P. S. Printable Superelastic Conductors with Extreme Stretchability and Robust Cycling Endurance Enabled by Liquid-Metal Particles. *Adv. Mater.*, 1706157 (2018).*

17. Lin, Y., Cooper, C., Wang, M., Adams, J. J., Genzer, J. & Dickey, M. D. *Handwritten, Soft Circuit Boards and Antennas Using Liquid Metal Nanoparticles. Small* 11, 6397-6403 (2015).”

And some other useful papers have also been added:

In page 17, “19. Deng, B. & Cheng, G. J. Pulsed laser modulated shock transition from liquid metal nanoparticles to mechanically and thermally robust solid-liquid patterns. Adv. Mater., e1807811 (2019).”

In page 19, “56. Jia, Y., Zhai, X., Fu, W., Liu, Y., Li, F. & Zhong, C. Surfactant-free emulsions stabilized by tempo-oxidized bacterial cellulose. Carbohydr. Polym. 151, 907-915 (2016).”

Responses to Reviewer #3:

The authors report an effect of “superiority in coalescing” of EGaIn droplets in presence of CNFs’ nanofibers. The obtained double sides free stand films possess relatively high mechanical properties, as well as high reflectivity, and actuating behavior responsive to humidity, voltage and photo.

However, my opinion the manuscript is written in strange manner, .i.e. it looks like more “advertising presentation” than the original scientific paper. I am looking to the root of the effect and want to understand its physics.

***Reply 3:** We would like to thank the reviewer for carefully reading the manuscript and kindly giving the comments. We hope that our answers below and the changes highlighted (blue for relocation and red for modification) in the manuscript could fully address the Reviewer’s concerns.*

The paper requires a major revision based on the comments shown below:

Title:

Better: Evaporative sintering?!

And may be better fibers, while fibrils may be also OK!

***Reply:** According to this specific suggestion, the title has been changed to “**Evaporation-induced sintering of liquid metal droplets with biological nanofibrils for flexible conductivity and responsive actuation**”.*

(a) Introduction is too long. The problem can be explained and formulated in two paragraphs (paragraphs 1 and 2 can be easily omitted, the rest shorten. Try to be very specific.

***Reply 3-a1:** That is a good question!*

Accordingly, we have reorganized and shortened the Introduction! And some sentences have been removed or relocated.

In this work, the presence of CNFs not only assisted to sinter the EGaIn droplets via the suspension evaporation, but also served as the structural stabilizer for EGaIn-CNFs composites (e.g., coatings and free-standing Janus films). So the authors think one task of this work is to present the phenomenon of evaporation-induced sintering and as well to discuss the possible underlying mechanism. And the other task is to fully reveal the functions and potential applications of the composites for the audience to exhibit the significance of the method of evaporation-induced

sintering. And these results have been demonstrated in the Results and Discussion sections. For the logical integrity of the paper, relevant background information has been introduced in the beginning of the Introduction section like an explicit statement for the close association between nanomaterial coalescence and flexible electronics (e.g., soft robotics and wearable devices).

However, the following specific issues should be addressed:

- For the general reader please explain the specific properties of the eutectic GaIn (EGaIn) alloy;

Reply 3-a2: We appreciate this suggestion!

The specific properties of EGaIn have been added in the revised manuscript as follows:

In page 3, “In contrast to rigid solid nanomaterials, droplets of liquid metal (LM, e.g., EGaIn with 75%-gallium and 25% indium⁸) show the superiority in sintering into integral liquid conductors which could withstand physical deformation as diverse as bending, twisting, stretching and compression”.

In Page 5, “EGaIn (with 75% gallium and 25% indium⁸), acting as a typical eutectic alloy (with a melting point ~ 15.8 °C), remains fluidic at room temperature with a large surface tension (624 mN m^{-1}) and high conductivity ($3.4 \times 10^6 \text{ S m}^{-1}$).”.

- Please describe the process of “natural evaporation”, as you understand it

Reply 3-a3: We appreciate this suggestion!

“Natural evaporation” means that the sintering process could be spontaneously accomplished under ambient conditions (e.g., room temperature of ~ 20 °C, ordinary pressure of ~ 0.1 MPa and any relative humidity when the solvents can evaporate).

Clearer descriptions of the process have been made, and the usage of this item has been carefully revised as follows:

In Page 4, “Herein we showed that **natural evaporation** could sinter colloidal suspensions of EGaIn droplets in the presence of biological nanofibers (NFs) as low as 0.05 wt% (NFs, with the diameter of ~ 5 – 10 nm) of cellulose” has been revised to “Herein we showed that **evaporation at the ambient condition** (room temperature of ~ 20 °C, ordinary pressure of ~ 0.1 MPa and relative humidity (RH)

of ~40%) could sinter colloidal suspensions of EGaIn droplets in the presence of biological nanofibrils (NFs, with the diameter of ~5–10 nm) as low as 0.05 wt% of cellulose,”.

*In Page 11, “During **natural evaporation** of solvents, meniscus-shaped capillary bridges would form among the colloids and yield an attraction force.” has been revised to “During **evaporation** of solvents, meniscus-shaped capillary bridges would form among the colloids and yield an attraction force.”.*

*In Page 14, “following by drying at ambient conditions (~0.1 MPa, 25 ± 3 °C, RH 40 ± 5%).” has been changed to “following by drying at ambient conditions (~0.1 MPa, 25 ± 3 °C, RH 40 ± 5%) **at least for one day for completely drying before further test.**”.*

- Please explain why the use of biological fibers is so important. For sure, the general statement on “a local pressure at nanoscale” does not directly related to the nature of the fibers.

Reply 3-a3: *Thanks for this question!*

Firstly, biological NFs are widely distributed in biomass, and in special, cellulose NFs (CNFs) are the most abundant biopolymer nanomaterial with an aspect ratio up to 10^2 and a large elastic modulus up to 10^2 GPa. So biological NFs have promising applications in flexible electronics (e.g., soft robotics and wearable devices) with advantages of low cost, biodegradability and sustainability. And in this work, these NFs could promote substrate adhesion of EGaIn layers, and serve as structural matrix of EGaIn droplets for free-standing, humidity responsivity, biocompatibility and biodegradation in electronic applications. Secondly, for the process of solvent evaporation-induced sintering biological NFs could not only attach on the surfaces of EGaIn droplets for lower droplet size and higher colloidal stability, but also help to rupture the encapsulating shells of EGaIn droplets through threefold contributions as shown in Fig. R3-1: (1) higher capillary force based on the Equation 3, (2) enhanced capillary force by spitting liquid bridges into small ones, and (3) pulling tension by NF contraction among the droplets during evaporation. Biological NFs typically have plentiful polar groups like carboxyl, so they can easily attach on the surface of EGaIn droplets for lower droplet size and lower contact angle, and thereby a higher EGaIn droplets for lower droplet size. Biological NFs have an aspect ratio up to 10^2 and a large elastic modulus up to 10^2 GPa, so they are ideal to produce spitting liquid bridges and pulling tension. According to this

specific comment, control experiments have been carried out using other polymers instead of biological NFs like biopolymer sodium alginate, negative surfactant sodium dodecyl benzene sulfonate, positive surfactant cetyl trimethyl ammonium bromide, and nonionic surfactant polyvinylpyrrolidone. At the same conditions (e.g., a concentration of 0.2 wt% in aqueous suspension, room temperature of ~20 °C, ordinary pressure of ~0.1 MPa and RH ~40%), they cannot sinter EGaIn droplets through solvent evaporation.

Fig. R3-1. Schematic illustration of sintering mechanism: Capillary force promoted by biological NFs; Local pulling tension induced by biological NFs; Splitting into multiple liquid bridges.

(b) Results. Figure 1, in the present form, may be moved to the supplementary file. Please do not use cartoons in presenting your data.

In Figure 1a please show the typical TEM image of the fibers, which was used to obtain the their diameter distribution and the corresponding distribution;

Figure 1b please show the typical SEM/TEM image that was used to obtain the droplets distribution before sonication and corresponding droplets size distribution; Figure 1 c: if you can, show the image of the suspension;

Figure 1d: show the nice image of the cross section for the obtained film.

Figure 1f: show the nice image of the cross section for the film formed without use of CNFs.

Figure 2: only conductivity data should be presented.

Reply 3-b: *We appreciate this comment very much!*

According to the reviewer's valuable suggestion, the Fig. 1 has been revised as shown in Fig. R3-2, and other images and cartoons have been removed into the Supplementary Information (Supplementary Figure 1) as shown in Fig. R3-3:

Fig. R3-2. **Evaporation-induced sintering of EGaIn droplets with biological NFs for free-standing films and coatings.** (a) Typical TEM image of CNFs. The inset gives optical image of nematic CNFs suspension. (b) Typical SEM image of EGaIn droplets produced by sonicating bulk EGaIn in CNF suspension. The top-right inset gives corresponding optical image. Sonication: 60 min; Concentration (ϕ_{CNF}): 0.2 wt%. (c) Schematic illustration of EGaIn droplet encapsulated in oxide shell and with CNFs attached on surface via interactions with Ga^{3+} . Diameter histogram of EGaIn droplets in Fig. 1b (Bottom left) and effect of ϕ_{CNF} on average EGaIn diameter after 60 min sonication (Bottom right) were given as the inset. (d) Evaporation-induced sintering into free-standing films (Top) with mirror-like bottom surface and grey top surface, and coatings (Bottom)

on different substrates through mask depositing, hand-writing or drop-casting. (e) Cross-sectional SEM images of coating layers of EGaIn droplets with (Top) and without CNFs (Bottom).

Fig. R3-3. (a) Schematic illustration of synthesizing biological NFs from biomass via liquid exfoliation. TEM images and diameter histogram of amyloid NFs (b & c) and silk NFs (d & e). The inset gives visual observation of corresponding aqueous suspensions (0.1 wt%).

Analysis of sintering mechanism in Fig. 2 has been moved to the Discussion section.

(c) Mechanism. Should be moved to the discussion part.

This is the important part of the paper. First, here you should explain in more details what does it means “suspension evaporation” and “evaporating sintering”.

Reply 3-c1: We sincerely appreciate your comments and the valuable suggestions!

Accordingly, the Mechanism section was removed to the Discussion part. “Suspension evaporation” means the evaporation of solvents (e.g., water) of the EGaIn droplets suspensions in the presence of

CNFs. While “*evaporating sintering*” means the solvent evaporation-induced coalescing and sintering of EGaIn droplets which should be mainly due to the enhanced capillary force from the CNFs. Corresponding revisions have been made as follow:

The title has been changed to “**Evaporation-induced** sintering of liquid metal droplets with biological nanofibrils for flexible conductivity and responsive actuation”.

In Page 2, “*evaporative sintering* of LM droplets under ambient conditions” have been changed to “**evaporation-induced sintering** of LM droplets under ambient conditions.”

And “Thus this *evaporative sintering* approach not only extends...” has been revised to “Thus this **evaporation-induced sintering** approach not only extends”.

In Page 7, “via the *suspension evaporation*, but also...” has been changed to “via **evaporation**, but also...”.

“shared a common bilayer structure (**Fig. 2a**), suggesting the presence of a precipitation separation process during the *suspension evaporation* (**Supplementary Figure 14 & 15**).” has been changed to “shared a common bilayer structure (**Fig. 1e**), suggesting the presence of a precipitation separation process during **the evaporation** (**Supplementary Figure 15-16**).”.

“during *evaporative sintering*. EGaIn droplets, their higher colloidal stability (**Supplementary Figure 14**).” has been changed into “during **evaporation-induced sintering** (**Supplementary Figure 17a**). EGaIn droplets, their higher colloidal stability (**Supplementary Figure 17-18**).”.

In Page 10, “This precipitation separation process seemed to be essential for *evaporative sintering*.” has been changed into “This precipitation separation process seemed to be essential for **sintering**.”.

In Page 20, “Fig. 1 **Evaporative** sintering EGaIn droplets” has been changed to “Fig. 1 **Evaporation-induced** sintering of EGaIn droplets”.

And items of “*evaporative sintering*” has been revised to “**evaporation-induced sintering**” in other part of the text.

Next:

- It is understandable that “Due to the high density (6.28 g mL^{-1}),¹⁶ EGaIn droplets precipitated first ... (with the layer thickness of $>5 \text{ }\mu\text{m}$). CNFs had relatively high colloidal stability and deposited on the EGaIn layer as the insulating layer.”
- However the following important statement is not well documented “CNFs might also exist within the conductive layer because of their presence on and between the EGaIn droplets during evaporative sintering. EGaIn droplets, when having the smaller size and attaching more CNFs, might also exist within the insulating layer because of their higher colloidal stability.”

Can you provide some evidences for the above: e.g. based on TEM studies of the film?

Reply 3-c2: *We appreciate this comment!*

Accordingly, the bottom EGaIn-rich layer was dissolved with 0.1 M HCl and washed carefully with pure water using centrifugation. The obtained product in the bottom of the tube was re-dispersed in pure water and observed by TEM. As shown in Fig. R3-4a, CNFs could be obviously observed with TEM. And cross-sectional SEM image of the top CNFs-rich layer was also observed, and some EGaIn droplets were found embedded in the CNFs-rich layer without sintering (Fig. R3-4b).

Fig. R3-4. (a) TEM image of the obtained product in the bottom of the centrifuge tube after the bottom EGaIn-rich layer was dissolved with 0.1 M HCl. (b) Cross-sectional SEM image of EGaIn droplets in CNFs-rich layer.

The corresponding results were added in Supplementary Figure 17 and Fig. 4a of the revised manuscript, respectively.

*In Page 7, “during **evaporative sintering**. EGaIn droplets, because of their higher colloidal stability (Supplementary Figure 14).” has been changed into “during **evaporation-induced sintering***

(Supplementary Figure 17a). EGaIn droplets, because of their higher colloidal stability (Supplementary Figure 17-18).”.

More important: Equation 3 does not contain directly any characteristics of the CNFs and calculated value of $F_{\text{Capillary}}$ could be present for the droplets themselves. It can be speculated that CNTs influence the wetting angle, but in this case, the wetting studies in droplets-CNFs system are critical. Bottom line, the Eq. (3) without additional discussion does not explain the observed effect.

Reply 3-c3: *It as a good question!*

The contact angle of the EGaIn droplet wrapped with CNFs was difficult to detect due to this ultra-small size in nano scale. So intrinsic contact angle of CNFs (5°) was used in the estimation according to the reference (Carbohydr. Polym. 2016, 151, 907). According to the reviewer’s specific suggestion, the capillary force without CNFs addition was also calculated. However, the contact angle of the EGaIn droplet coated by Ga_2O_3 was not found in the literatures. So the contact angle of bulk EGaIn coated by Ga_2O_3 was detected by the authors ($\sim 86^\circ$) and the capillary force was evaluated about 0.9 MPa much lower than that of EGaIn droplet wrapped with CNFs (~ 13.2 MPa).

And the corresponding revision has been added in the revised manuscript as follows:

In Page 11, “While only a pressure of 0.9 MPa was obtained in absence of CNFs (see detailed calculation in Supporting Information).”.

In Page 4 of the Supporting Information, “Without the presence of CNFs, if the contact angle of bulk EGaIn with Ga_2O_3 surfaces ($\sim 86^\circ$) was used, the capillary force was calculated as 0.9 MPa, which is much lower than the critical sintering pressure ~ 5 MPa.^{S6}” was added.

Next, re “spitting the liquid bridges”. The claim is”...a portion of CNFs would infiltrate into the gaps among the EGaIn droplets, and sinter them during the following evaporation...” It is completely speculation. Can you provide any evidences for this mechanism? (see above comments to the more detailed characterization of the obtained films). Here you may use some nice “cartoon”-type figure to show all mentioned parameters and processes.

Reply 3-c4: *We appreciate this suggestion!*

According to this comment, the cross-sectional SEM images at different time periods during solvent evaporation were observed. Before test, the specimen was frozen in liquid nitrogen ($-196\text{ }^{\circ}\text{C}$) for 15 min and dried in a freeze-drier (FD-1C-50, Beijing BoYiKang) at $-50\text{ }^{\circ}\text{C}$. As shown in Fig. R3-5, the process of solvent evaporation and sintering of the EGaIn droplets was much clearer. During solvent evaporation, the relatively heavy EGaIn droplets with a density of $\sim 6\text{ g cm}^{-3}$ (should be also wrapped by amount of CNFs) was apt to precipitate first on the bottom, while the settlement of free CNFs with a relatively light density of $\sim 1\text{ g cm}^{-3}$ were much more difficult due to its colloidal stability (Supplementary Figure 3, 8 and 9). With further evaporation, the solvent became less enough and meniscus would emerge within the EGaIn droplets-CNFs stacks. At the same time part of the free CNFs in the suspension would infiltrate into the gaps among the EGaIn droplets with the solvent owing to capillary and gravitational forces. When the capillary force was high enough to rupture and sinter the EGaIn droplets, a conductive EGaIn layer formed with a little CNFs embedded. And excessive CNFs that could not infiltrate into the EGaIn gaps were dried and covered on the surface of the EGaIn layer to form a CNFs-rich layer. To be noted, some EGaIn droplets having the smaller size or attaching more CNFs also have a low settling speed, so they were embedded in the CNFs-rich layer without sintering.

Fig. R3-5. (a) Cross-sectional SEM images of the film at different time periods during solvent evaporation. (b) Zoom-in images of Fig. R3-5a for the bottom part of the film at the last one hour during evaporation.

And the obtained SEM images have been added into the revised manuscript as follows:

In Page 7, “shared a common bilayer structure (**Fig. 2a**), suggesting the presence of a precipitation separation process during the **suspension evaporation** (**Supplementary Figure 14 & 15**).” has been changed to “shared a common bilayer structure (**Fig. 1e**), suggesting the presence of a precipitation separation process during **the evaporation** (**Supplementary Figure 15-16**).”.

“during **evaporative sintering**. EGaIn droplets,... because of their higher colloidal stability (**Supplementary Figure 14**).” has been changed into “during **evaporation-induced sintering** (**Supplementary Figure 17a**). EGaIn droplets, because of their higher colloidal stability (**Supplementary Figure 17-18**).”.

In Page 14 of the Supplementary Information, “**Supplementary Figure 18. (a) Conductivity variation versus time during drying suspension of EGaIn droplets on glass. Suspension volume of EGaIn droplets: 0.5 mL; $\phi_{\text{CNF}} \leq 0.2$ wt%; Glass substrate: $1.5 \times 1 \text{ cm}^2$. Before solvent evaporation, only a low ionic conductivity was detected in the fluid layer of EGaIn droplets. During solvent evaporation, the capillary force may be high enough to sinter EGaIn droplets and produce a conductivity increase. (b) Cross-sectional SEM image when drying suspension of EGaIn droplets on glass for indicated periods of time. Each specimen underwent a freeze-drying process before SEM characterization. (c) SEM images for characterization of sintering process of EGaIn droplets during solvent evaporation. Before solvent evaporation, EGaIn droplets was apt to precipitate first on the bottom due to its large density of $\sim 6 \text{ g cm}^{-3}$, while free CNFs which were not attached on EGaIn droplets tended to keep colloiddally stable in the suspension (Supplementary Figure 3, 8 and 9). At the final stage of solvent evaporation, capillary force was high enough to sinter EGaIn droplets and formed a conductive EGaIn-rich layer. Part of CNFs would stay over the EGaIn-rich layer and dry into a CNFs-rich layer. To be noted, part of EGaIn droplets may had the smaller size and attach a large amount of CNFs, which would show higher colloidal stability and thus stay in the CNFs-rich layer after completely drying.**” has been added.

(d) Properties. Please explain what is so special in that “The bottom bright layer reflected light in a way akin to bulk LM with a reflection $\geq 80\%$ within the visible-light region (400–720 nm)”.

Reply 3-d: That is a good question!

Bright luster of metals like gold may be not special. But here the bright layer of LM was obtained by solvent evaporation-induced sintering of EGaIn nanodroplets. To fully coalescing and sintering EGaIn droplets by a natural and spontaneous process was unique and miraculous. On the other hand, the optical property of the film was tested in order to show the Janus properties of the resultant film. One face is bright metal while another is lackluster biomass, which may be special for soft yet robust films.

*So the reflective property of the two face of the film was given in one sentence contrastively as “The bottom bright layer reflected light in a way akin to bulk LM with a reflection $\geq 80\%$ within the visible-light region (400–720 nm), whilst the top CNF layer displayed high light absorption and thus a low reflection $\leq 20\%$ (**Fig. 2a and Supplementary Figure 20**).” in Page 7 of the manuscript.*

REVIEWERS' COMMENTS:

Reviewer #1 (Remarks to the Author):

The authors have addressed my comments, and I can recommend the work to be published in its current form.

Reviewer #2 (Remarks to the Author):

The authors have adequately addressed my comments. My only remaining concern is regarding Fig. S7 -- I find Fig. R2-3 more useful in demonstrating the binding of CNF to the liquid metal particles than what is currently shown in Fig. S7.

Reviewer #3 (Remarks to the Author):

Authors did a decent work addressing the reviewer's comments.
Revised version of the manuscript can be recommended for publication